# Recent Advances and Perceptive Insights into Powder-Mixed Dielectric Fluid of EDM

**DOI:** 10.3390/mi11080754

**Published:** 2020-07-31

**Authors:** Asarudheen Abdudeen, Jaber E. Abu Qudeiri, Ansar Kareem, Thanveer Ahammed, Aiman Ziout

**Affiliations:** Mechanical Engineering Department, College of Engineering, United Arab Emirates University, Al Ain 15551, UAE; 201990133@uaeu.ac.ae (A.A.); 201990209@uaeu.ac.ae (A.K.); 201870099@uaeu.ac.ae (T.A.); ziout@uaeu.ac.ae (A.Z.)

**Keywords:** EDM, SR, TWR, PMEDM, MRR

## Abstract

Electrical discharge machining (EDM) is an advanced machining method which removes metal by a series of recurring electrical discharges between an electrode and a conductive workpiece, submerged in a dielectric fluid. Even though EDM techniques are widely used to cut hard materials, low efficiency and high tool wear remain remarkable challenges in this process. Various studies, such as mixing different powders to dielectric fluids, are progressing to improve their efficiency. This paper reviews advances in the powder-mixed EDM process. Furthermore, studies about various powders used for the process and its comparison are carried out. This review looks at the objectives of achieving a more efficient metal removal rate, reduction in tool wear, and improved surface quality of the powder-mixed EDM process. Moreover, this paper helps researchers select suitable powders which are exhibiting better results and identifying different aspects of powder-mixed dielectric fluid of EDM.

## 1. Introduction 

Among all the non-conventional machining processes, electrical discharge machining (EDM) is one of the major and popular machining processes. Electrical discharge machining is also known as spark machining, spark eroding, die sinking, and wire burning or wire erosion. This technique is usually used to machine hard materials and high-temperature resistant alloys which are electrically conductive. The principle of the EDM technique is the use of thermoelectric energy to erode conductive components through rapidly occurring sparks between the uncontacted electrode and the workpiece [1]. It is an electro-thermal non-traditional machining process, where electrical energy is used to generate electrical spark, and the material is removed from the workpiece by a series of rapidly requiring current discharges between two electrodes separated by a dielectric liquid. It can be used to machine difficult geometries in small batches or even on a job-shop basis [2]. A large body of research discusses EDM processes with different objectives [3,4,5,6,7,8,9,10,11].

In the case of development of super-tough electrically conductive materials like carbides, stainless steels, hastalloys, nitralloys, waspalloy, nomonics, etc., the demand for non-traditional manufacturing processes has become more relevant. These super tough materials, which have extensive applications, such as manufacturing of dies, automobile, and aerospace components, are very difficult to machine by conventional methods. To machine all the electrically conductive materials irrespective of their hardness and toughness, EDM processes extensively use thermal energy [2].

In non-conventional machining processes, EDM has tremendous potential on account of its versatility because of its applications in modern industries. The EDM process can also produce holes, external shapes, profiles or cavities in an electrically conductive work piece by means of controlled application of high-frequency electrical discharges by vaporizing or melting the work piece material in a particular area. The electrical discharges are the results of controlled pulses of direct current and occur between the tool electrode (cathode) and the work piece (anode) [12]. Generally, one of the largest uses of the EDM process is in tool-, die-, and mold-making [13]. In the powder-mixed EDM (PMEDM) process, electrically conductive powder is used in dielectric medium which reduces the insulating strength of dielectric fluid, and it increases the spark gap between tool and work piece which will result in an improvement of material removal rate (MRR) and surface finish [14,15]. The mostly used powders are aluminum, chromium, graphite, silicon, copper, and silicon carbide. Pulse on time, duty cycle, peak current, and concentration of the powder added to the dielectric fluid of EDM are the main variables to study the process of the performance in terms of MRR and surface roughness (SR) [16].

Material removal rate is defined as the volume of material removed over a unit period [17,18]. It is commonly expressed by the unit (mm^3^/min). A high value of the discharge voltage, peak current, pulse duration, duty cycle, and the low values of pulse interval will result in a higher MRR. In addition to these abovementioned electrical parameters, non-electrical parameters and material properties also have their own significant influence on the MRR.

Tool wear ratio (TWR) can be defined as the ration of volume of materials removed from the tool electrode to that of work piece. It can be also called as electrode wear ratio (EWR). Tool wear ratio depends on electrode polarity and the electrode materials properties. Surface roughness is a very important parameter to consider in die-sinking EDM. In most die-sinking operations, separate finishing and roughing operations are carried out to complete the final product. It is represented by the average SR and measured in microns [19].

The aim of this paper was to provide comprehensive information and details of the PMEDM process. It is pertinent that there is great importance given to the review articles to bring out intricacies, parameter–response co-relations, and critical analysis of its response for better utilization of this process. This paper gives state-of-the-art current research studies conducted with PMEDM variants for machining different materials.

The paper is organized to begin with a brief introduction of PMEDM, its variants and introduces its working principle. It further discusses the optimization of process parameters and the types of EDM studies for different powder-mixed dielectrics in various EDM processes. Moreover, performance measures in PMEDM and materials are also discussed. Furthermore, effect of powders and their concentration, magnetic field-assisted PMEDM, modeling and simulation of PMEDM, combined and hybrid process with powder-mixed dielectric fluid are scrutinized in the paper.

In addition, other objective functions, such as analysis of performance of plain water, performance of water mixed with organic compounds, performance of commercial water-based dielectrics and their surface effects are also evaluated. The main text of this article provides a meticulous review on major areas of PMEDM research on different powder materials. It may help researchers find suitable dielectric fluid having better properties according to their needs. The last section of the paper draws conclusions and trends of the reviewed bodies of research are subsequently drawn.

## 2. Different Types of EDM Techniques

There are many types of EDM. This research categorized EDM processes under three main categories, namely, Sinker EDM, Wire EDM, and fast hole drilling EDM. Spark machining’s working principle is based on the erosion of the material by frequent sparks between the workpiece and the device or tool that is immersed in a dielectric medium. A gap separates the workpiece and the electrode to establish a pulsed spark through which the dielectric fluid flows. Schematic representation of the basic working principle of EDM process is shown in Figure 1.

### 2.1. Wire EDM

One of the most emerging non-conventional manufacturing processes is the wire electrical discharge machining [20,21]. Widely this process is used to machine hard materials and intricate shapes which are not possible with conventional machining methods. It is more efficient and economical. In wire EDM, an electric spark is created between an electrode and the work piece [22,23]. The spark is the visible evidence of the flow of electricity. This electric spark creates intense heat of temperature ranges from 8000 to 12,000 degree Celsius, which melts almost anything. The spark is carefully controlled and localized so that it only affects the surface of the materials. The EDM process does not affect the heat treat below the surface of the work piece. In wire EDM, the spark always takes place in the dielectric of de-ionized water. The conductivity of the water is carefully controlled for making an excellent environment for the EDM process. Here, the water acts as a coolant and flushes away the eroded metal particles [24]. The working principle of wire EDM is shown in Figure 2.

The wire EDM process has a wide range of applications such as in die making, electronics and automotive industries [25,26].

As per the literature, the main parameters are:Pulse-on time (Ton)Pulse-off time (Toff)Servo voltage (V)Peak current (I)Gap voltage (Vgap)Dielectric flow rateWire feed rateWire tension

### 2.2. Sinker EDM

It is also called cavity-type EDM or volume EDM. It consists of an electrode and work piece submerged in an insulating liquid, such as, more typically, oil, or, less frequently, other dielectric fluids. The electrode and work piece are connected to a suitable power supply [27]. The power supply generates an electrical voltage between the two parts. As the electrode approaches the work piece, the dielectric breakdown occurs in the fluid, which forms a plasma channel and small spark jumps. These jumping sparks usually strike on one at a time. The sinker EDM process uses an electrically charged electrode that is configured to a specific geometry to burn the electrode’s geometry into a metal component. This process is commonly used in dies and tool manufacturing [28]. A schematic diagram of a sinker EDM is shown in Figure 3.

Here, the main components are power supply, dielectric system, electrode, and the servo system. So, this is the schematic which explains the principle of sparking. The work piece is usually connected to the positive terminal of the power supply. Here, the shape of the tool is different, and the same shape will be replicated as well on the work piece. The sparking takes place through the different zones or the through different points which are closer to work piece. Due to the fact of this sparking, the plasma formation zone will create the bubbles and high pressure which subsequently collapse and erode the work piece. Whenever the sparking takes place, erosion of the work piece will also occur. Thus, we have seen in the case of EDM, small erosion will take place on the electrode as well. Sinker EDM is one of the advanced methods for machining electrically conductive materials [29,30,31,32,33].

### 2.3. Fast Hole Drilling EDM

Figure 4 shows fast hole drilling EDM. This EDM was designed for fast, accurate, and deep hole manufacturing. By concept, it is similar to sinker EDM process but the electrode is a rotating tube conveying pressurized jet of dielectric fluid. It can make deep holes in about a minute, and it is a good way to machine holes in materials which is too hard for twist drilling machining. This EDM drilling process is mostly used in aerospace industries for producing cooling holes in aero blades and other components which requires cooling. It is also used in industrial gas turbine blades, in molds and dies, and in bearings.

The abovementioned EDM methods was about machining using dielectric fluid. Various powders can be mixed to improve their efficiency.

## 3. Powder-Mixed EDM 

To improve the ionization and ease in the frequency of spark between the cathode (tool) and work piece, fine-grained conductive powder is added into dielectric fluid in the EDM process [34,35]. In the PMEDM process, the suitable material in the powder form is mixed into the dielectric fluid tank [36] for the better circulation of the dielectric fluid by the mixing system which includes aluminum, graphite, copper, chromium, silicon carbide, etc. The presence of this powder makes the process mechanism substantially different from the conventional EDM process [37,38]. Here a spark gap was provided by the additive particle which ranged in between 25–50 micrometers. The voltage applied was in between 80 and 320V, and the electrical field ranged in between 105–105 V/m [39]. A large body of research discussed the PMEDM processes with different objectives [40,41,42,43,44,45,46,47,48,49,50].

### Principle of Powder-Mixed EDM

The influence of the powder in the PMEDM mainly depends on the powder parameters, that is, powder material, particle size, and particle concentration. There are many powder materials mixed with dielectric fluid including aluminum (20%), chromium (0.9%), graphite (5%), silicon (0.03%) or silicon carbide (0.3%). The previous studies used partial sizes of 10–25 μm, also the particle concentrations are about 6 g/L. These fine powders mixed to the dielectric fluid in the EDM, resulted in the reduced insulating strength and the generation of more spark [24]. Thus, table MRR and surface quality can be expected. The powder particles are energized and moved in crisscross directions. Under the action of applied electric force, these particles arrange in a suitable form at different locations in the sparking area [51]. Tool steel, alloy steel, and especially nickel-based super alloy Inconel-800 has been commonly used as workpieces by various researchers [45,52,53]. Other commonly used workpieces in PMEDM are H-11 Die Steel [54], AISI D2 Die Steel [41], Hastelloy [55], H 13 steel [56], W300 Die Steel [57], EN 31 Steel [58], AISI H-11 [59], Stavax [60], etc. The schematic representation of the principle of PMEDM is shown in Figure 5.

One of the main issues faced while researching PMEDM is that the literatures available on this topic is limited. However, it can be easily found from the available literature that considerable efforts have been directed to improve the quality of the surface by suspending the powder particles in the dielectric fluid used in the EDM. There were few works reported for the improvement of machining efficiency of powder-mixed EDM.

By the addition of powder particles to the electric discharge machining process, the dielectric fluid will modify some process variables, and it creates the condition to achieve a higher surface quality. Here, the improvement in the polishing performance of conventional EDM and the analysis were carried out by varying the silicon powder concentration and flushing flow rate over a set of different process areas. It was found that, effects in the final surface after machining was evaluated by surface morphologic analysis and measured through some surface quality indicators [61]. Jeswani (1981) [62] conducted experiments with the addition of 4 gm/L of fine graphite powder in the dielectric and found that MRR improved by 60% and the electrode wear ratio reduced by about 15%. And the further investigations show high efficiency application of water mixture and glycerin for reducing disadvantages in EDM sinking using pure water.

Purna Chandra Sekhar et al. [40] explained elaborately about their research work. That saying, the material removal mechanism of the EDM process was very complex, and also the theoretical modeling of this process was very difficult. Powder-mixed dielectric is a promising research area. Only a few studies have touched on the introduction of using nano powders into EDM process. The author reported that most of the research work carried out with aluminum, silicon, and graphite powders [63,64,65,66] and some with other types of powders. They are Cr, Ni, Mo, Ti, etc. [67,68,69]. The Impact of such machining on MRR, SR, and TWR was studied by the most available research works on the powder-mixed dielectric fluids [40]. Generally, the tool electrodes used in PMEDM can be copper, tungsten, brass, aluminum, and graphite [46,60,70,71,72,73].

When cryotreated electrode was used, MRR, TWR, and SR decreased by 12%, 24%, and 13.3%, respectively, and when SiC powder was used, MRR increased by 23.2% and TWR and Ra decreased by about 25% and 14.2%, respectively [74]. Moreover, the machining performance can be improved using graphene nanofluid as the dielectric [75]. Studies on plasma channel expansion, nano-powder-mixed sinking and milling in micro-EDM were also carried out [76,77].

## 4. Various Powders Used in EDM

Powders mixed with dielectric fluid in EDM can be categorized into six different types, namely, aluminum, silicon, chromium, graphite, silicon carbide, and nickel micro powder. Each of these types has its own characteristics which makes it suitable to be used for different machining conditions. In order to meet the required conditions, these powders have many properties such as MRR, SR, TWR, etc. General composition of widely and recently used powder types (materials) for EDM process considered for each type are summarized in Table 1 and their physical properties are shown in Table 2.

The material removal rate can be increased by mixing powder with the dielectric fluid as compared with ordinary EDM process. Also, it can be increased by the increasing of the peak current. Peak current in a higher value will produce a rougher surface in the EDM process. That means different mixing powders in different input parameters have different responses and results. Review of optimization of process parameters are given in Table 3.

## 5. Relevant Studies on PMEDM

In this section, the studies dealing with the PMEDM process are reviewed based on the following classification scheme:

Type of EDM process: The relevant studies classified according to the EDM processes used powders mixed with the dielectric fluid are as follows:D-S—Die sinking EDM;WEDM—Wire EDM;FHD—Fast hole drilling.

From the authors’ perspective, reviewing the literature related to the PMEDM technique shows that most of the studies discussed the powder-mixed for D-S EDM technologies such as Reference [85] While few studies discussed the powder mixed for wire cut such as Reference [16], and for fast hole drilling there are very few researchers studying the effect of PMEDM and its process parameters [86]. But there are relevant researches available related with PMEDM using other EDM processes or by modifying the above three processes [87].

Objective functions:

The following objective functions are reported in the current relevant literature:Objective 1: Performance measures in PMEDM;Objective 2: Powder materials;Objective 3: Effect of powders and their concentration;Objective 4: Magnetic field-assisted PMEDM;Objective 5: Other objective functions.

Using this classification scheme, Table 4 chronologically lists the studies for powder-mixed dielectric fluid in different EDM process.

### 5.1. Performance Measures in PMEDM

One of the important performance measures in EDM is MRR. Many research studies explored a number of ways to improve the SR and MRR in EDM. In order to obtain an optimum combination of performance measures for different work–tool interfaces, in major bulk of research studies introduced the optimization of process parameters. The reports generally agree that the SR has decreased and improved surface finish, with lower pulsed current and pulse-on time values and relatively higher pulse-off time. High-quality SR cannot be achieved when pulses of long duration are used during finishing process [107]. Several researchers tried innovative ways to get better performance parameters such as MRR improvement as well. Previous studies focused on reducing the TWR because the wear of the electrode tool affects the electrode tool profile and leads to a lower precision [108,109,110]. While referring the available literature references on the process, a need is felt to summarize all the results and conclusions made by different researchers [19]. There are four techniques to improve performance variables:By electrode design;By controlling process parameters;By EDM variations;By powder-mixed dielectric.

The effect of first three techniques mentioned above can be found in [1,12,75]. This paper concentrates on the effect of powder-mixed dielectric on the performance variables. Khan et al. [89] used a work piece of stainless-steel AISI 304 material and an electrode of copper material, they found that addition of 0.3 mg/L of Al_2_O_3_ in dielectric fluid will results in the improvement of SR and the MRR will increase. Here, the electrical discharge seems to be broken into many small discharges due to the presence of powder, which will result in the formation of many small cavities with smaller depths of the surface of the work piece. This produces smoother surface. As the result of the increasing of Ton, the heat energy is absorbed during a longer time making the surface less stressed that produces a better surface. In addition, the increase of MRR results in the increasing of pulse on time.

A Kumar et al. [111] studied the performance of EDM process for machining Inconel 825 alloy by mixing Al_2_O_3_ nanopowder in deionized water. The experimental investigation revealed that by setting optimal combinations of process parameters, the maximum MRR of 47 mg/min and minimum SR of 1.487 μm will be, respectively, 44 and 51 percent higher compared to conventional EDM process. S Patel et.al says that rotary tool is a recent innovation of PMEDM. Authors used aluminum oxide (Al_2_O_3_) powder with a particle concentration of 0.5–1.5 gm/L into the dielectric to study improved machining performance and found tremendous changes [81]. Hence, it can be noted that performance can be improved by adding various powders in dielectric fluids [112,113].

### 5.2. Powder Materials 

In the PMEDM, the dielectric fluid mixed with the powder of different materials. Thus, the floating particles impede the burning process by creating a higher discharge probability and lowering the breakdown strength of the insulating dielectric fluid [101]. Material removal rate, SR, increase and reduces the TWR value along with the improvement of sparking efficiency and the gap distance enlarged by the conductive powders and dispersed the discharges more randomly throughout the surface. Here the micro cracks are reduced and the thickness of the recast layer becomes smaller. Thus, the corrosion resistance of the machined surface may subsequently improve [45].

From the study of surface modification technique using EDM with TiN-mixed fluid. A Muttamara and J Mesee [102] introduced a surface modification technique using EDM with TiN-mixed fluid. The study came to some points that the re-solidified layer containing TiCN can be generated on EDMed surface. The authors revealed that, the surface after EDM with TiN-mixed kerosene is rougher than that in EDM with kerosene-type fluid. The modified layer thickness grew up to 57 micrometers. On this thick layer there were few micro cracks and voids detected. Hardness of the modified surface with TiN mixed kerosene came to 980 HV which is harder than that EDMed with pure kerosene. On the machined surface, a hard layer containing TiCN can be formed. Smaller hardness value with a little higher SR value can be obtained by mixing TiN powder with dielectric kerosene [102].

For the machining of Inconel 718 (Nickel-based super alloys), copper tungsten in cylindrical shape was used as electrode tool. Here the nano alumina is used as the additive dielectric. It is used because of its high thermal conductivity. Due to the nano alumina having high thermal conductivity, more heat is distributed and dissipated to the surface of work piece to limit the size of craters produced. Mainly nano alumina has combined the effects of low electrical resistivity and low density. Low electrical resistivity creates a high spark and high thermal conductivity takes more heat away [70,114].

For the deposition in the powder-suspended EDM, the concentration of powder in the gap between the electrode and the work piece must be very high. To stabilize the machining process, the electrode moves reciprocally. Due to the reciprocating motion of thick electrode, the powder is flushed from the gap. During the downward motion of the electrode, the powder concentration is low, whereas in the conventional powder-suspended EDM, the powder hardly adheres on the work piece. To keep the powder concentration high, then powder should always provide into the gap [68]. Most researchers used concentrations of powder below 20 g/L and powders of micro-size ranging from 1 to 55 μm or powders of nano-size ranging from 20 to 150 nm [38].

Nano powder-mixed EDM is one of the recent advancements in this field. Certain studies were conducted for finding various properties of adding nano particles in the process for improving the efficiency [17,115,116,117,118,119]. Similarly experiments using graphene oxide flakes and scrolls are carried out for achieving related objectives [120].

### 5.3. Effect of Powders and Their Concentration

The concentration of powder-mixed with the dielectric fluid also discussed in many studies. For example, Sugunakar et.al [91] found that the MRR increases with the mixing of different powder particles into the dielectric medium. Due to the increasing of graphite powder from 0 to 9 g/L, the MRR further increases. Thereby, MRR decrease with the supplementary addition of graphite powder from 9 to 14 g/L, whereas insignificant increase in MRR is noticed when increasing in Al powder from 0 to 405 g/L and it decrease with increase in Al powder from 4.5 to 9 g/L. Here, we can observe that a considerable increase in MRR with increase in Al powder substance from 9 to 14 g/L. However significant increase in MRR was observed with increasing in combination of Al and graphite powders (1:1 ratio) from 0 to 4.5 g/L and then it decreases further increasing in combination of Al and graphite powders (1:1 ratio) from 4.5 to 14 g/L [91].

By improving material removal rate and optimizing various machining parameters in EDM, Vivek Kumar and Prakash Kumar made some summary points. They noted that MRR of electric discharge machining can be improved using optimization of various factors which are discharge Current, Ton, Toff and voltage. Discharge current has most significant parameters for MRR. Moderately, material removal rate effected by voltage and Ton time. Here the MRR significantly influenced by Toff [87].

The material removal rate is low at low current values as the discharge current is increased, the MRR also increases. The pulse time increases as MRR increases [92]. All factors have significant effect in varying degrees on the EDM performance. Pulse current is one of the most significant factors affecting the material removal rate, dimensional accuracy, and surface integrity of drilled hole. Comparatively the better electrode material is copper, because it gives better surface finish, high MRR, and less electrode wear than Al [121]. As per the optimization of process parameters of surfactant and graphite powder-mixed dielectric EDM through Taguchi–Grey relational analysis, it was found that the significance of each process parameter in sequence is, peak current. It is the most significant factor than surfactant concentration and graphite powder concentration based on largest delta (Δ) value which was found from the test response. The optimal combination of these process parameters was obtained [122]. According to Lamichhane et al. [123] the momentous parameters enhancing MRR (maximum 19.01 mg/minimum) were I, Ton along with the addition of hydroxyapatite nanoparticles (HAp) nano-powder in the dielectric fluid. Adding HAp also improves SR (0.340 µm) of machined surface. Hence, it shows the smallest particles produced the best surface finish while increasing the recast layer thickness. Suitable particle concentration tends to efficiency of the process and improved stability. Shinde et al. [124] used aluminum, silicon, and silicon carbide fine abrasive powders with particle concentration of 2 gm/L and size of 44 m were added into the SEO25 (spark erosion oil) dielectric liquid of electrical discharge machine. The finest concentration for maximizing the performance depends on powder characteristics.

### 5.4. Magnetic Field Assisted PMEDM

Magnetic field was found to be more effective in case of lower current values for desirable MRR and TWR. Here, the magnetic field resulted in an increase in the OC with strengths at different levels of current. For the SR, magnetic field resulted in the deterioration in surface finish for higher current settings, while it resulted in better surface finish at low current. It enhances the surface hardness by the support of magnetic field strength [103]. In another research, Al–metal matrix composites (MMCs) hybrid ED machining was studied in the magnetic field integrated in the traditional EDM. The experimental results witnessed a decrease in the micro hardness values and a decrease in recast layer thickness, followed by a major effect on MRR and surface finish when machining higher spark energy in the magnetic field. The experimental results brought consistency to the method and an excellent correspondence with experimental verifications [125]. The input processing parameters, magnetic field strength, pulse-on/off length, peak current, electrode variant, and workpiece were evaluated to determine their after-effects on the microhardness (MH) response and recast layer formation during Al–SiC composite machining. The experimental results indicate a 22% decline in surface microhardness and a thinner recast layer formation in the magnetic field coupled with higher spark energy [126]. Thus, a higher spark energy formed along with the magnetic field can decline the surface hardness. Bains et al. [126] studied the individual effect of machining parameters, namely, peak current, pulse on time, pulse off time, powder concentration, and magnetic field on material removal rate and tool wear rate. The effect of peak current on material removal rate and tool wear rate, followed by pulse on time, concentration of powder, and magnetic field was found to be dominant [127].

### 5.5. Other Objective Functions 

Modeling and simulation of PMEDM process was also discussed. For the simulation process, a work piece model was generated using a numerical simulation like ANSYS software. This model was considered an AISI D2 die steel and the tool material was copper. The most common tool used is AISI D2 steel for the EDM applications, and it is best machined by copper electrode. Using finite element method, the model was tested under various conditions. For this required chemical composition, mechanical properties and the thermal properties of the material were given. For the analysis, process parameters for PMEDM like voltage, current, heat input to work piece, radius of spark, pulse duration, pulse on time, types of flushing, polarity, powder type and size, frequency constant, powder concentration, electrode lift, tool electrode diameter are considered. And the FEM model is validated by comparing it with the predicted theoretical results with experimental data. 

The model developed for the study of PMEDM, can be further used to obtain the distributions of the temperature over the work piece, distributions of residual stress, flow of metal, and also to find out the cracks on the PMEDM work piece. So, the model can be used as an industrial tool to predict the evolution of temperature, stress, strain on the machined work piece [42].

A number of studies were performed using models and simulation approaches to achieve specific objective functions such as one investigation [128] which reported the development of a piezoelectric servo system to improve the gap control in dry EDM. In another investigation [129], an optimization system was proposed to generate the optimum EDM process parameters. Han et al. [130] used a simulated method developed for wire EDM. Additionally, Yadav et al. [131] developed a model to reflect the EDM process’s thermal stress and Tsai et al. [132] a semi-empirical model for surface finishing of machined workpieces was developed in EDM. Furthermore, Wang et al. [133] developed a semi-empirical MRR model in the EDM process. Williams and Rajurkar [134] created an EDM wire process model which was developed to research the characteristics of the machined surfaces. Cogun and Savsar [135] introduced a statistical model for studying time-lapse variations in the EDM process.

Salah et al. [136] conducted numerical modelling of the temperature distribution caused by the EDM process. From these results the MRR and total roughness were deduced and compared with experimental observations for stainless-steel type AISI316L. Some other researchers, such as Gao et al. [137], experimented in such a way that a vibration model was set for the workpiece or the tool (wire). Srivastava and Pandey [138] concentrated on kerf analysis in micro-WEDM and lateral wire vibration and breakdown distance determined for a micro-WEDM process. They also advanced a lateral vibration model for the wire which they used to measure the maximum wire vibration amplitude. These authors also conducted numerous experiments with different machining parameters on stainless steel. Rafał Swiercz et al. [139] says PMEDM with reduced graphene oxide flakes causes more stable electrical discharges with less energy and resulted in a uniform thickness of the recast layer. Studies using graphenes are recent in this field. Different characteristics like erosion characteristics, surface characteristics, surface integrity, and machining parameters were studied using grapheme oxide [140,141,142].

Also, the PMEDM was combined and hybrid with other processes as presented in Reference [56]. The aim of this combination was to control the process stability and to compare the pulse shapes in different processes. There was an electronic circuit connected to a computer display which recorded the current and voltage pulses. In this hybrid machining mode, the tool electrode will vibrate by ultrasonic frequency and SiO_2_ nano powder was added to dielectric fluid. Due to the application of ultrasonic vibration on the tool electrode, an ultrasonic head of 20 kHz frequency of vibration and 20 W power mounted on EDM machine were obtained. Properties of typical dielectrics used in PMEDM are given in the Table 5.

Using more advanced dielectric fluids and powders in EDM, the MRR and other properties of the machined materials will improve, and it makes the machining process very simple [98]. Besides that, health and atmosphere effects should be considered. The use of eco-friendly dielectric fluid will help to minimize the harmful environmental effects. Also, the use of water-based dielectric in die sink EDM influence the following factors [94].

Performance of plain water;Performance of water mixed with organic compounds;Performance of commercial water-based dielectrics;Surface effects.

Effect on productivity by using deionized water or tap water may result in higher levels of material removal rate in some special cases like, in the time of using of brass electrode at negative polarity. In the effect on surface integrity, the deionized water usually has oxides on the machined surface and lower values of SR, while hydrocarbon oil has a contaminated appearance with carbon atoms inside the craters [95]. Most of the sinking processes use hydro-carbon oil like kerosene-based oil as the working fluids. Kerosene-based working fluids are used considering other factors such as health, safety, and environment. However, kerosene is inflammable and therefore undesirable, as the possibility of fire hazard has always been of great concern in sinking EDM [96]. To overcome the pollution and harmful vapor throughout the EDM machining process using the hydro-carbon dielectric, the EDM process is further refined and a novel method was proposed. Its name was given as dry-EDM (DEDM). In this process, air or compressed is passed through a thin walled tubular pipe, whenever the compressed gas cools the inter electrode gap and relived the debris from the machining zone [97].

Earlier studies also assessed the surface integrity and stresses of machined tool steel surfaces after EDM processes. For example, in one study [143], the impact of EDM process parameters on the 3D surface topography of tool steel was explored, while in another study [144], the crystallographic and metallurgical properties of the white layer in the Böhler W300 ferrite steel EDM were documented. In another investigation [145], residual stress was measured in material SKD11. Effect of EDM machining parameters on the surface characteristics and damage to machining of AISI D2 tool steel material [146] was studied. Other works surface alterations of tool steel were studied during the EDM process for AISI H13 and AISI D6 [147,148]. Some studies [149] looked at the effect of process parameters on the fatigue life of the AISI D6 steel tool. Changes in the chemical composition of the re-solidified layers of the electrodes and workpieces were also studied during the T215Cr12 die steel EDM [3]. Several experiments centered on resulting residual stresses in the machined surface, for example, Reference [150].

Both MRR and EWR decrease with increase in the temperature. Within the higher temperature, the droplets can align as much as easier under the action of the electrostatic force due to the smaller viscosity of the emulsion. And the better flushing condition can slightly improve the EDM performance. However, the improvement of the flushing condition cannot improve further performance of the EDM [151]. When hydro-carbon oil-based working fluid was used, there were several aerosols or gases generated due to the decomposition of the hydro-carbon oil. Besides the emission in the air, the accumulation of decomposition substance in the working fluid also increases the process complexity of waste fluid itself [99,100].

In consideration of accretion EDM mechanism, titanium powder suspended working oil is used as the dielectric. Here the titanium carbide will deposit when both discharge energy and power density are small. But, in the case of discharge power density exceeding the critical power density for the removal of carbon steel, then it melts and will evaporate at low discharge energy. Due to the consequent process, the titanium powder was not deposited, and the carbon steel was simply removed from the removal region [152]. The experimental setup is shown in Figure 6.

Besides its machining purpose, according to Furutani et al. [47] EDM is also used for the deposition process. In EDM the working fluid suspended with silicon powder will increase the SR because of the dispersion of discharge decreases the frictional coefficient. In EDM with MoS_2_ suspended in the working oil, the MoS_2_ could be deposited on the metals which have low melting point than MoS_2_. Dispersion of the lubricant duration occurs due to the uneven concentration of MoS2. Gap length expansion caused the improvement of roughness [104].

## 6. Discussion

Reviewing the publication related to dielectric fluid in the EDM process showed that the majority of the studies investigated the effect of variation of powder concentration in different process parameters, such as voltage, current, pulse rate, pulse on time, and it has been observed that for low current values the MRR is very low. But the MRR increased with the increase in the discharge current. For reverse polarity, the spark energy was low at low discharge current and pulse on time. Other bodies of researches have been conducted for improving the machining efficiency and for reducing the tool wear ratio. These studies point out the advanced methods for achieving better surface finish, reducing the tool wear ratio, and increasing MRR. Moreover, this method has tremendous application in the field of mold and die making and nano size hole drilling.

PMEDM is really useful for:Machining of widely available advanced materials like MMCs, insulating ceramics like TiO_2_ which have been successfully machined by dispersing various powders into EDM dielectric;Production and machining of micro products and sophisticated micro mechanical elements.

But, the setup has the following drawbacks too:Electrically conductive material can only be machined;For achieving more accuracy in machining, it consumes more time;Health problems are occurring due to the usage of some dielectric fluids;Materials like water hardened die steel, molybdenum high-speed tool steel have not been tried as work material.

## 7. Conclusions

The reviews on the state of art, studies on the dielectric fluid in EDM processes lead to the following conclusions.

According to the general agreement on the results, different conductive powders, with different concentrations can improve the material removal rate in EDMPeak current, pulse on and off time, duty cycle, voltage, discharge current, tool angle, powder concentration, nozzle flushing, and grain are the input parameters that improve the material removal rate and changing of SR with the different configuration of the above input parametersAccording to the major observations by the researchers, the increase in SR and improvement in MRR were observed in the mixing of powders into the dielectric mediumThis review reveals that, with a specific increase in concentration of powder in the dielectric fluid, the MRR and SR will increase. Increasing the concentration of powder in the dielectric fluid beyond the certain optimum concentration of particles in the dielectric, short-circuit discharges occur, and, as a consequence, MRR decreases. From this analysis, it can be concluded that PMEDM holds a brilliant promise in the application of EDM, in particular, regarding process efficiency and work piece surface quality. Therefore, an extensive study is required to understand the machining mechanics and other aspects of PMEDM and will be performed as future research.

## Figures and Tables

**Figure 1 micromachines-11-00754-f001:**
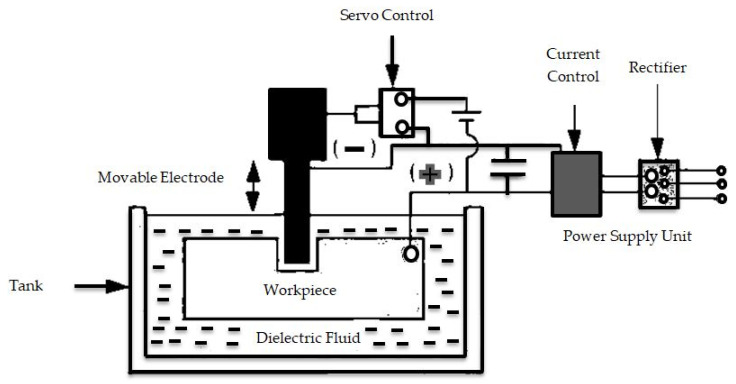
Schematic representation of the basic working principle of electrical discharge machining (EDM) process.

**Figure 2 micromachines-11-00754-f002:**
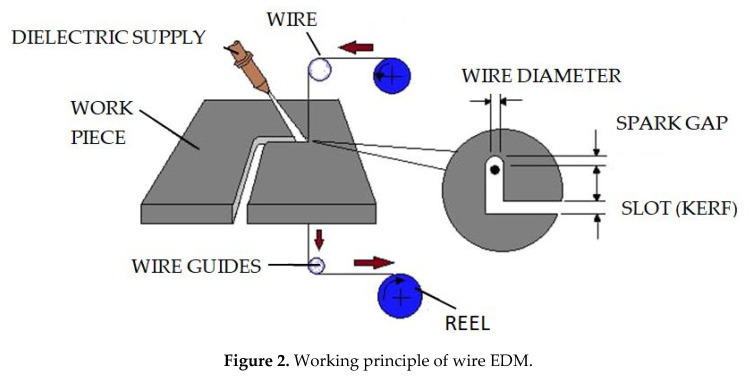
Working principle of wire EDM.

**Figure 3 micromachines-11-00754-f003:**
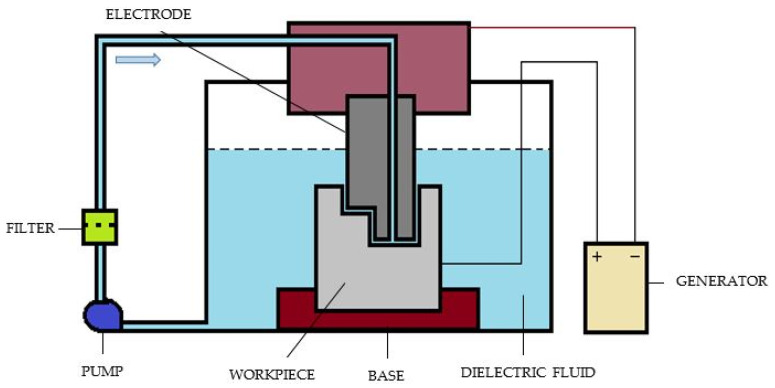
Schematic diagram of sinker EDM.

**Figure 4 micromachines-11-00754-f004:**
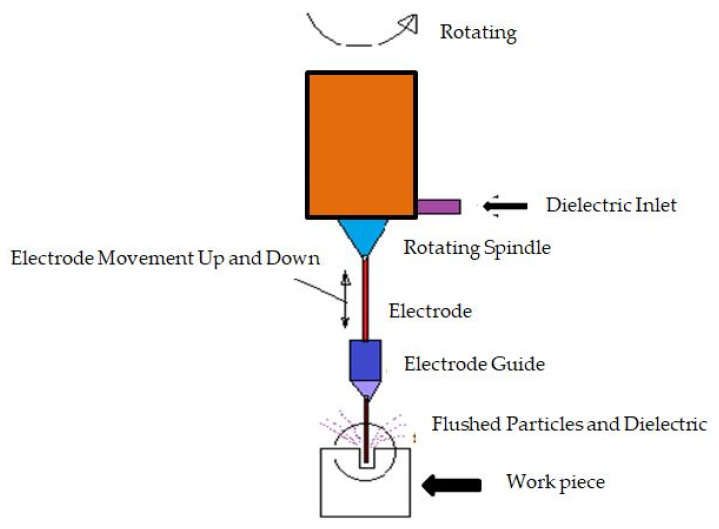
Fast hole drilling EDM.

**Figure 5 micromachines-11-00754-f005:**
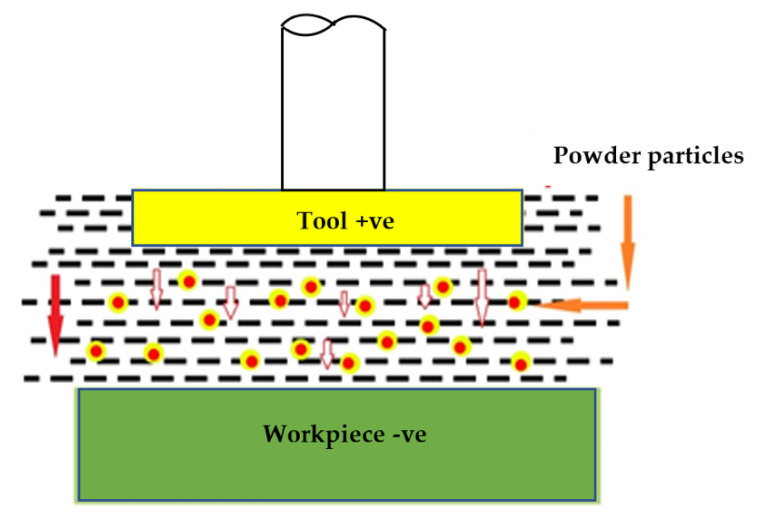
Schematic representation of the principle of powder-mixed EDM.

**Figure 6 micromachines-11-00754-f006:**
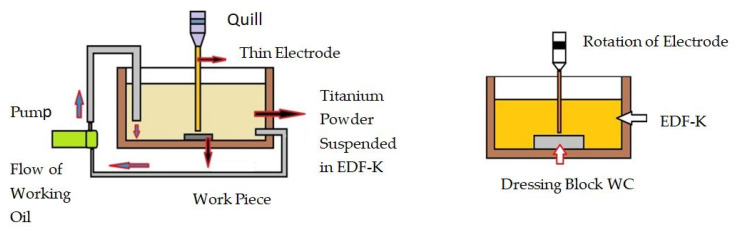
Experimental setup for accretion mechanism.

**Table 1 micromachines-11-00754-t001:** Details of general composition of various powders used by researchers for different EDM processes.

Powder Type	Composition (weight %)	Parameters Studied
Silicon	0.03	material removal rate (MRR), surface roughness (SR) [70]
Aluminum	20	MRR, SR, Tool wear ratio (TWR) [78]
Silicon	0.3	MRR [41]
Chromium	9	MRR, TWR [69]
Nickel micro powder	6	MRR [79]

**Table 2 micromachines-11-00754-t002:** Commonly used powders in powder-mixed EDM (PMEDM) and their physical properties [80].

Material	Density (g/cm^3^)	Electrical Resistivity (μΩ-cm)	Thermal Conductivity (W/m-K)
Aluminum (Al)	2.70	2.89	236
Graphite C	1.26	103	3000
Chromium (Cr)	7.16	2.6	95
Copper (Cu)	8.96	1.71	401
Silicon (Si)	2.33	2325	168
Nickel (Ni)	8.91	9.5	94
Silicon Carbide (SiC)	3.22	1013	300
Titanium (Ti)	4.72	47	22
Tungsten (W)	19.25	5.3	182
Alumina (Al_2_O_3_)	3.98	103	25.1
Boron Carbide (B_4_C)	2.52	5.5 × 105	27.9
Carbon nano tubes (CNTs)	2.0	50	4000
Molybdenum Disulfide (MoS_2_)	5.06	106	138

**Table 3 micromachines-11-00754-t003:** Review on optimization of process parameters.

Work Material	Powder	Input Parameter	Response Variables	Optimization Technique	Results	Reference
En-31	Silicon	Peak current, Pulse on, Duty cycle, Powder concentration	MRR, SR	Response surface methodology (RSM)	Powder concentration and peak current were the most influential parameters	[70]
Inconel 718	Aluminum	Voltage, Discharge current, Duty cycle, Powder concentration	MRR, SE, WR	One variable at a time	Size and particle concentration significantly affect machining efficiency	[81]
AISI-D2 Die steel	Silicon	Peak current, Pulse on time, Pulse off time, Powder concentration, Grain, nozzle flushing	MR	Taguchi method	The peak current and concentration of silicon powder mostly influences the machining rate	[41]
EN-8	Chromium	Current, Tool angle, Powder concentration, Duty cycle	MRR, TWR	RSM	The most significant parameters affecting MRR are powder concentration and current, whereas, current and electrode angle greatly influences TWR	[69]
EN-19	Nickel micro powder	Peak current, Duty cycle, Electrode angle, Powder concentration	MRR, TWR	RSM	ANOVA results revealed that the current was the most dominant factor affecting both MRR and TWR increases with increase in current and powder concentration	[79]
AISI 1045 Steel	Aluminum	Current, Voltage, Pulse on time, Duty factor constant	MRR, SR	Taguchi method	As the concentration of aluminum powder and grain size in EDM oil increases, surface roughness starts decreasing. MRR and surface roughness are equally important. With the increase in concentration of aluminum powder and grain size MRR and surface finish of AISI 1045 Steel increases	[63]
W300 Die Steel	Aluminum	Peak current, Pulse on time, Powder concentration, and polarity	MRR, EWR, SR, WLT	Signal-to-noise (S/N) ratio and the analysis of variance (ANOVA)	Polarity plays an important role in PMEDM. High MRR is obtained in positive polarity, whereas better surface quality (surface roughness and white layer thickness) is achieved in negative polarity. Distilled water can be used as dielectric fluid instead of hydrocarbon oil and, moreover, the performance can be improved by the addition of aluminum powder	[57]
EN 31 Steel	Silicon	Pulse on time, Duty cycle and Peak current, Powder material, Powder size, Powder concentration, Dielectric type, Peak voltage, Pulse off time, Polarity, Inter electrode gap (IEG)	MRR, TWR, WR, SR	RSM	MRR and SR roughness have been measured for each setting. The use of powder-mixed dielectric promotes the reduction of surface roughness and enhances material removal rate	[80]
SKD-11	Aluminum chromium copper and silicon carbide powders concentration	Pulse on time, Peak current	MRR, TWR, SR	RSM	The discharge gap distance and material removal rate increased as powder granularity was increased. Aluminum produced the largest discharge gap enlargement and silicon carbide produced the smallest.	[82]
AISI D2 Die steel	Chromium	Peak current, Pulse on time, Pulse off time, Powder concentration	MRR, TWR, SR	Taguchi, Anova	With the increase in current and pulse-on time, MRR increases. Due to the increased concentration of chromium powder, MRR tends to decrease. TWR is mainly affected by current. With the increase in current, TWR increases. Also, TWR tends to decrease with the increase in chromium powder concentration. surface roughness is higher with the increase in pulse-off time	[83]
AISI D3 Die Steel	Aluminum Powder	Peak current, Pulse on time	MRR, TWR, SR	Central composite design (CCD) of response surface methodology (RSM)	Maximum MRR is obtained at a high peak current of 14 Amp, higher Ton of 150 μs, and high concentration of Al powder 6 g/L. Low TWR and SR are made with low peak current of 2 Amp, lower ton of 50 μs and higher concentration of Al powder of 6 g/L.	[84]

**Table 4 micromachines-11-00754-t004:** List of the studies for different powder-mixed dielectric in various EDM processes.

No.	Author, Year	EDM Process				Objectives				
		D-S EDM	W EDM	FHD EDM	Other EDM Processes	1	2	3	4	5
1	[2]				x	√	√			
2	[55]				x	√	√	√		
3	[16]		x			√	√			
4	[88]		x							√
5	[51]	x	x			√				
6	[61]				x	√	√	√		√
7	[62]	x				√	√	√		
8	[40]					√	√	√		√
9	[70]	x				√	√			
10	[78]				x	√	√	√		√
11	[41]	x					√	√		
12	[69]				x	√	√	√		
13	[79]				x	√	√	√		
14	[19]				x					√
15	[89]				x			√		
16	[86]			x						√
17	[90]				x		√	√		
18	[91]	x				√	√	√		
19	[87]				x					√
20	[92]				x	√	√	√		
21	[93]				x					√
22	[37]				x		√	√		√
23	[94]				x		√	√		
24	[95]				x					√
25	[96]	x								√
26	[97]				x					√
27	[98]	x								√
28	[99]	x								
29	[100]				x					√
30	[101]				x		√	√		√
31	[45]				x	√		√		√
32	[102]				x	√	√	√		
33	[12]				x		√			√
34	[68]				x					√
35	[103]				x					√
36	[47]				x					√
37	[104]				x		√			
38	[105]				x					√
39	[40]				x	√	√	√		√
40	[42]				x			√		
41	[106]				x	√				
42	[17]	x								√

**Table 5 micromachines-11-00754-t005:** Properties of typical dielectrics used in PMEDM.

Dielectric Name	Specific Heat (J/kg-K)	Thermal Conductivity (W/m-K)	Breakdown Strength (kV/mm)	Flash Point (°C)
Deionized water	4200	0.62	65–70	Not applicable
Kerosene	2100	0.14	24	37–65
Mineral oil	1860	0.13	10–15	160
Silicon oil	1510	0.15	10–15	300

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
