# Peer review of "Recent Advances and Perceptive Insights into Powder-Mixed Dielectric Fluid of EDM"

_micromachines, 2020, doi:10.3390/mi11080754_

Round 1
Reviewer 1 Report
This paper (' Recent Advances and Perceptive Insights into Powder Mixed Dielectric Fluid of EDM ' for Micromachines) is mainly about the Electrical Discharge Machining (EDM). The paper is good to be accepted. However, there are still something need to modify in this paper. My detailed comments are as follows:
- There are some spelling mistakes, for example, in line110 should be pulse-off time or line 124 Figure 3 with a space or figure 4: electrode movement…etc.
- Line 208 which is table 3, there are some format error. The underline should remove. So does table 5.
- From line 220-222 should be stated in table.
- The words in the figures should be the forms as the manuscript.
- In conclusion, there is a punctuation mistake. Please modified it.
- The English need to be rewritten.
Reviewer 2 Report
The paper refers to the review of recent advances and perspectives in the PMEDM. The topic is quite interesting but requires major revisions before it can be accepted for publication for the following reasons:
- The title and abstract of the article do not much to the content. Authors write that the “paper reviews about the recent advances in the powder mixed EDM process”, however for 124 cited references over 54 had more than ten years old.
- The physical analysis of PMEDM in the relation of the literature review is poor, please see “State of the art in powder mixed dielectric for EDM applications” Precision Engineering, Volume 46, October 2016, Pages 11-33, https://doi.org/10.1016/j.precisioneng.2016.05.010,
- Authors present in the article different types of EDM (Sinking EDM, WEDM, EDM Drill), with suggestions of use PM to this technology. However, the analyses of using PM are limited only to objective functions presented in the articles without the comment of results for WEDM and Drill EDM
- There is no analysis of the literature review of the latest PMEDM research, including, for example, using nanoparticles of graphene, reduced graphene oxide in the dielectric.
- Some conclusions are vague and may mislead readers, for example, a statement “With the increase in the concentration of powder in the dielectric fluid the MRR and SR will increase is true only for some specific conditions. There are many examples of results in the literature that after exceeding the certain optimum concentration of particles in the dielectric, short-circuit discharges occur and, as a consequence, MRR decreases.
- Each of the Figure (1-6) should be correct: resolutions and description should be improved, besides this:
- Figure 1. Basic working principle of EDM process – the figure is too simple, the scheme of current and voltage waveforms are wrong,
- Figure 3. Diagram of sinker EDM – the figure is too simple: instead battery it should be generator; authors write: So, this is the schematic which explains the principle of sparking – without current and voltage waveforms this scheme does not explain the principle of sparking, in sinker EDM the dielectric in most cases is not given to the gap through the electrode,
- Figure 5. Principle of powder mixed EDM - the figure is too simple; not represent PMEDM principle, not refer to discharge propagation through additional particles, no physical analysis of the process
